# Flexible Pressure Sensor Array with Multi-Channel Wireless Readout Chip

**DOI:** 10.3390/s22103934

**Published:** 2022-05-23

**Authors:** Haohan Wangxu, Liangjian Lyu, Hengchang Bi, Xing Wu

**Affiliations:** In Situ Devices Center, School of Communication and Electronic Engineering, East China Normal University, 500 Dongchuan Road, Shanghai 200241, China; 10192100496@stu.ecnu.edu.cn (H.W.); hcbi@cee.ecnu.edu.cn (H.B.)

**Keywords:** flexible sensor array, resistive pressure sensor, readout chip, ballistocardiogram, pulse wave

## Abstract

Flexible sensor arrays are widely used for wearable physiological signal recording applications. A high density sensor array requires the signal readout to be compatible with multiple channels. This paper presents a highly-integrated remote health monitoring system integrating a flexible pressure sensor array with a multi-channel wireless readout chip. The custom-designed chip features 64 voltage readout channels, a power management unit, and a wireless transceiver. The whole chip fabricated in a 65 nm complementary metal-oxide-semiconductor (CMOS) process occupies 3.7 × 3.7 mm^2^, and the core blocks consume 2.3 mW from a 1 V supply in the wireless recording mode. The proposed multi-channel system is validated by measuring the ballistocardiogram (BCG) and pulse wave, which paves the way for future portable remote human physiological signals monitoring devices.

## 1. Introduction

Flexible electronic devices are widely used in detecting human physiological signals for their advantages of being attachable and wearable [1,2,3,4,5]. As a part of advanced devices, flexible sensors with complex sensing functions are significant in wearable devices [6,7].

Among the sensors, the flexible pressure sensors, which imitate the human sensory system, are widely applicated in wearable devices [8,9,10]. Flexible pressure sensors could convert subtle mechanical signals to electronic signals, mainly divided into piezoelectric, capacitive, and piezoresistive types. Compared with the other two kinds of flexible pressure sensors, piezoresistive pressure sensors have a simpler structure and more comprehensive measurement range [11]. Recently, piezoresistive pressure sensors have great application potential in health monitoring, artificial intelligence, and human posture detection [12,13,14,15].

The combination of sensors and near-sensor chips reduces communication power consumption [16]. The specifically designed sensor signal readout chip has the advantages of small size and low power, which are required for wearable devices. It is necessary to further study the integration of the flexible sensor array with a multi-channel signal processing chip. Gao et al. reported a sensor array integrated with flexible circuit for in situ perspiration analysis [17]. Recently, this method was applied to piezoresistive pressure sensors [18]. Researchers reported the development of a piezoresistive sensor array integrated with an ultrahigh-efficiency chip, which is the first to use a low-power artificial intelligence (AI) chip to realize human gait monitoring.

This paper proposes a remote health monitoring system to acquire ballistocardiogram (BCG) and pulse waves, which integrates a flexible piezoresistive pressure sensor array and a wireless multi-channel voltage readout chip. We also proposed a nonlinear resistance readout circuit to calibrate the inversely proportional resistive response of the pressure sensor.

## 2. Materials and Methods

Figure 1 shows the diagram of the proposed remote health monitoring system, which includes a flexible pressure sensor array (RP-C7.6LT-LF2, Legact Technology Co., Ltd., Shenzhen, China) and a custom-designed wireless multi-channel voltage readout chip. Carbon-based film is used as the pressure-sensitive layer. It is a porous structure. The scanning electron microscope (SEM) image (shot by Zeiss Gemini 450, Oberkochen, Germany) in Figure 1 shows the carbon particles and porous structure of the sensor. As the sensor is extruded, more current conductive paths are created leading a lower resistance. The multi-channel readout chip acquires the resistance values and transmits them wirelessly [19]. Such sensor array data could be used for BCG signals detecting.

The pressure sensor performance is studied. Figure 2a shows the characteristic testing system of the flexible pressure sensor, which is composed of a high precision tensile machine (UH6502, Youhong, Shanghai, China), an electrochemical workstation (CHI660D, CH Instruments, Shanghai, China). The tensile machine applies a time-varying force to the sensor and the electrochemical workstation measures its resistance change over time. The measured performance without post-processing shows the intrinsic characteristics of the sensor, which are shown in Figure 2b–d. Figure 2b illustrates the resistance change of the flexible pressure sensor with pressure increasing from 50 to 600 kPa. For all samples, resistance is inversely proportional to pressure, which can be described as
(1)$$\frac{1}{R}=6.8\times 10^{-6}\times P-0.0002$$
where R is the sensor resistance in kΩ, P is the applied pressure in kPa, and the goodness of fit is *R*^2^ = 0.99. Figure 2c depicts the change rate and sensitivity of the sensor by averaging five groups of samples with pressure increasing from 50 to 600 kPa. Sensitivity $S_{P}$ is defined as
(2)$$S_{P}=\left|\frac{\Delta R}{R_{0}}\right|\times \frac{1}{P}$$
where $\left|\Delta R/R_{0}\right|$ is the change rate of resistance. As the fitting curve shows, $S_{P}$ decreases with pressure, which is consistent with the inversely proportional resistance response. Figure 2d depicts the sensor output current change as the pressure increases stepwise from 200 to 700 kPa in units of 100 kPa. The current responses immediately to the pressure change almost with no delay, which reveals good repeatability at a wide range of pressure. The lower delay time leads to faster response, which can be used to track pressure changes induced by physiological signals.

The electronic signal from the pressure sensors is collected and processed. In our sensor system, the relationship between resistance and voltage is nonlinear. A specific chip is needed to process such signal. Figure 3a shows that the sensor resistance is converted into voltage change by connecting a fixed-value resistor in serial to form a resistor divider. Each node in the sensor array is independently connected to a switch and a divider resistor to avoid crosstalk between different channels. Only the sensor node being measured is turned on to generate a pressure-depended output voltage signal, while the others are disconnected from the bias voltage. The output voltage $V_{\mathrm{out}}$ can be described as:(3)$$V_{\mathrm{out}}=\frac{R_{\mathrm{ref}}}{R_{\mathrm{sen}}+R_{\mathrm{ref}}}\times V_{\mathrm{in}}$$
where $V_{\mathrm{in}}$ is the input voltage to bias the divider, $R_{\mathrm{ref}}$ is the resistance of the reference resistor, and $R_{\mathrm{sen}}$ is the resistance of the resistive pressure sensor. When $R_{\mathrm{ref}}$ is smaller than $R_{\mathrm{sen}}$, $V_{\mathrm{out}}$ can be approximated as
(4)$$V_{\mathrm{out}}\approx \frac{R_{\mathrm{ref}}}{R_{\mathrm{sen}}}\times V_{\mathrm{in}}$$

The inversely proportional resistance-to-voltage converter could be used for nonlinearity correction. Figure 3b shows that the sensor resistance is also approximately inversely proportional to the pressure. The proposed resistor divider makes a linear relationship between the output voltage and the pressure.

The sensitivity of the whole signal path, including the sensor and readout circuit, is obtained by taking Equation (1) into Equation (4). Assuming that the pressure measurement range is from 50 kPa to 1000 kPa, the sensitivity curves of the signal path of the linear and inversely proportional readout are shown in Figure 3b. As the pressure generated by the self-weight of the human body can generate a wide range of pressures, the nonlinear signal readout circuit can obtain higher sensitivity than the conventional linear circuits with a proportional voltage-resistance response.

The minimum identifiable input change $\Delta V_{\mathrm{in}}$ in the sensor readout analog front-end can be expressed as:(5)$$\Delta V_{\mathrm{in}}=\frac{V_{\mathrm{ref}}}{2^{N}\times S}$$
where S is the sensitivity of the signal path, $V_{\mathrm{ref}}$ is the reference voltage of the ADC, and N is the resolution of the ADC. Higher sensitivity S reduces the accuracy requirements of ADC to obtain the exact signal resolution, thus lowering system cost and power consumption. The linear relationship between the output voltage and pressure using the nonlinearity correction method also simplifies the post-reconstruction process from voltage to pressure, which reduces the power consumption of digital signal processing [20,21].

To further reduce the size and power of the proposed system, we utilize a self-designed 64-channel signal readout system on chip (SoC) to process the voltage-to-digital conversion and data transmission. As shown in Figure 4a, this SoC is composed of 64 analog front-end (AFE) channels, a data processing unit (DPU), a power management unit (PMU), and a wireless transceiver supporting radio frequency (RF) power harvesting. Each channel of the AFE includes a low noise amplifier (LNA) and a programmable gain amplifier (PGA), which are used to amplify the voltage signals from the sensor array shown in Figure 3a. A 10-bit SAR ADC digitizes every 16 channels at a sample rate of 40 kSps through time-division multiplexing. The DPU controls each block through on-chip registers and packages the output of ADCs into data frames. The frame packet is then sent through an on-off keying (OOK) transmitter operating at 3.2 GHz. The PMU includes a 13.56 MHz RF power harvester and a switched-capacitor DC-DC converter for wireless and battery-powered scenarios. On-chip low drop-out regulators (LDOs) and bias generators provide suitable operation voltage for each block [22,23]. Figure 4b shows the micrograph of the chip fabricated in a 65 nm complementary metal-oxide-semiconductor (CMOS) process with a total area of 3.7 × 3.7 mm^2^. Figure 4c summarizes the key specifications of the chip.

This chip supports dual-mode communication. In the wireless mode, the chip core consumes 2.28 mW from a 1 V supply, and the power distribution is shown in Figure 5a. The wireless data transmitter consumes most power, including the RF phase-locked loop (PLL) and the power amplifier (PA). Figure 5b shows the measured power distribution of the chip core in the wired-transmission mode. The total power consumption decreases to 0.48 mW. Figure 5c depicts the gain and input-referred noise (IRN) of the AFE. The measured mid-band gain of the AFE is 55 dB, and the −3 dB bandwidth is from sub-Hz to 10 kHz. The measured IRN is less than 8 μVrms integrating over the whole signal bandwidth. The chip can operate from a single supply ranging from 1.3 to 4.2 V. This chip also exhibits a higher power supply rejection ratio (PSRR) than the other work [24,25,26,27]. The replica-biasing scheme in the LNA guarantees a stable bias current despite high power supply interference without consuming much extra power and area [23]. The performance comparison with other works is shown in Figure 5d. The specifications of this chip are suitable for multi-mode physiological signal recording. The customized design has the advantage of low power and high channel count compared to commercially available chips.

## 3. Results

### 3.1. Subtle BCG Detection

We set up a high-density sensor array which is compatible with multi-channel signal readout. Here we show a case study to collect and process human physiological signals. BCG is an indicator of human heart activity. As shown in Figure 6a, the weight of a person sitting still naturally exerts a certain pressure on the sole, which is mainly distributed on the heel and forefoot. The sparse distribution of the artery at the heel makes it difficult to detect week BCG signal. The plantar artery near the big toe is densely distributed. Therefore, the forefoot is the best position to place the pressure sensor for BCG recording.

The sensor signal is amplified and recorded by the signal processing circuit described in Section 2. The recorded BCG signal is filtered by a second-order Butterworth low-pass infinite impulse response (IIR) filter with a cut-off frequency of 30 Hz. Figure 6b shows the obtained BCG signal. The weak BCG signal is vulnerable to foot shaking when measuring the signal. It is worth noting that the BCG signal is the micro-motion signal generated by the heart. The subject should keep the foot stationary to obtain a stable BCG signal. There are mainly three peaks in the BCG signal. The I-wave in the BCG signal is the minimum before the global maximum in the corresponding period, J-wave is the global maximum in the period, and K-wave is the global minimum after J-wave. Although the plantar pressure variations of the sensor lead to different detected BCG signals, the periodicity and characteristics of the BCG signal are still maintained after normalization. The representative peak of the I, J, and K wave could be distinguished clearly in Figure 6b.

### 3.2. Multi-Channel Pulse Wave Detection

The pressure sensor array can also be used to monitor the pulse wave during sleep. As shown in Figure 6c, the wrist squeezed between the bed and other objects provides a natural pressure, which could be used to detect the pulse wave at the radial artery. Considering the small area available for detection at the radial artery, a four-channel sensor array is attached to the wrist with a distribution shown in Figure 6c. Maintaining posture during sleep is challenging, an inflatable cuff is added to apply pressure to the wrist. The radial pulse waves measured at four different positions with the proposed wireless readout system simultaneously are shown in Figure 6d. According to the shape and amplitude of the pulse wave of each channel, the pressure distribution and pulse intensity at each point of the wrist can be judged. The amplitude of the pulse wave in each channel can be interpreted as a measure of distance between the sensor and the artery. The sensor closer to the artery would generate a pulse wave with a larger amplitude. In this work, the most significant pulse waveform amplitude appears at channel 2. Sensor arrays can also be used to measure the pulse wave transit time (PTT) and pulse wave velocity (PWV) [28]. In this work, the maximum distance between sensors in the array is 2 cm. As shown in Figure 6c, the pulse wave transmits from the proximal end (channel 2 and channel 4) to the distal end (channel 1 and channel 3). The sensor of channel 4 first senses the pulse signal, and the PTT can be calculated based on the foot-to-foot method [29]. The average PTT from channel 4 to channel 3 is 0.0865 s, and the average PTT from channel 2 to channel 1 is 0.0635 s. These could be further used for continuous noninvasive blood pressure monitoring.

## 4. Discussion

The flexible pressure sensor array, together with the custom-designed multi-channel signal readout chip provides a low-power solution to wearable health monitoring devices. The nonlinear resistance readout circuit calibrates the inversely proportional characteristic of the pressure sensor. The high linearity between output voltage and pressure simplifies the reconstruction process compared to traditional designs. This method is suitable for applications that need to directly record the pressure change. The nonlinear resistance-to-voltage conversion is easy to implement by simply decreasing the reference resistance in the resistor divider. The readout circuit becomes more sensitive at high-pressure conditions, common in BCG applications using bodyweight to provide pressure.

The robustness of physiological signal recording in a moving human body is important in wearable devices. As the size of each sensor used in this paper is small compared to the curvature radius of the wrist and heel, the bending of the foot and the wrist causes little sensor deformation. Meanwhile, the good bending reliability of the flexible sensor leads to little signal distortion. The proposed non-linear readout scheme is not limited to the specific material used in the paper, but can also be applied to other resistive sensors with similar inversely proportional responses.

Recent studies exhibit the feasibility of the cardiovascular (CV) parameters monitoring based on BCG. The amplitude between the second major wave and the third major wave is called J-K amplitude, which is an indicator of diastolic and systolic pressure [30]. Multivariate linear regression analysis of amplitudes and time intervals shows a relevance between BCG and CV parameters [31].

## 5. Conclusions

This paper reports a remote health monitoring system based on a pressure sensor array with a wireless multiple channels sensor signal processing chip. The system includes a flexible resistive pressure sensor array, a data transmission module, and a display module. The system can detect BCG signals and multi-channel pulse waves in sitting, sleeping, and other life scenes and exhibits great potential in human health monitoring and medical diagnostics. The proposed system has the potential to record from much more sensor sites, leading to a more precise measurement.

## Figures and Tables

**Figure 1 sensors-22-03934-f001:**
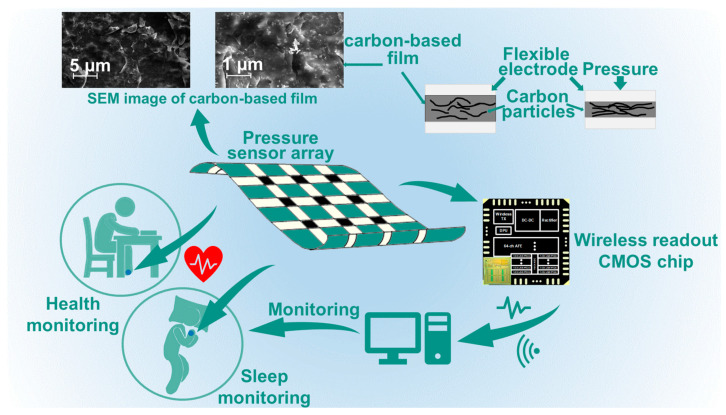
Overview of the remote health monitoring system with a flexible resistive pressure sensor array and a wireless multi-channel signal readout chip.

**Figure 2 sensors-22-03934-f002:**
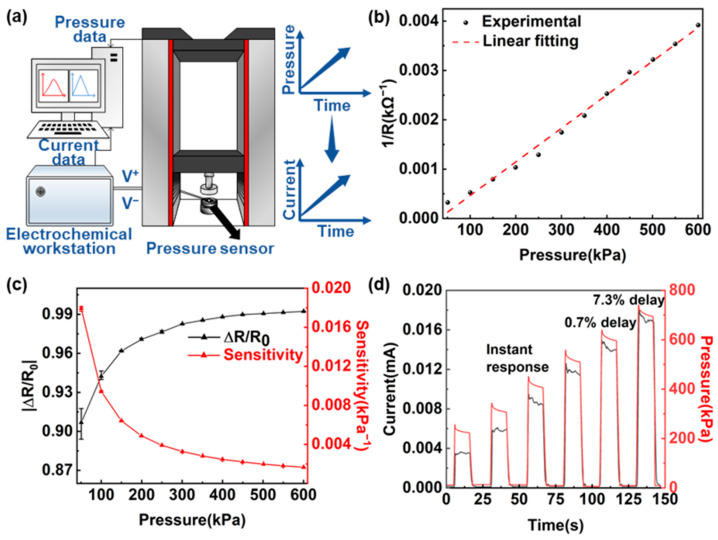
Characteristic test of the flexible pressure sensor. (**a**) The typical testing system of the flexible pressure sensor. (**b**) Changes in the reciprocal of resistance as a function of applied pressure. (**c**) Variations of deformation and sensitivity of five groups of samples with increasing pressure from 50 to 600 kPa. (**d**) The sensor output current changes with the step pressure, indicating its fast response.

**Figure 3 sensors-22-03934-f003:**
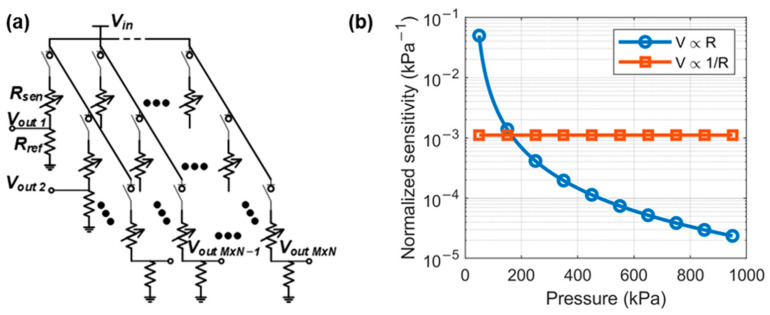
(**a**) Circuit diagram of the multi-channel resistance-to-voltage converter. (**b**) Sensitivity curves of the conventional linear and proposed nonlinear signal readout.

**Figure 4 sensors-22-03934-f004:**
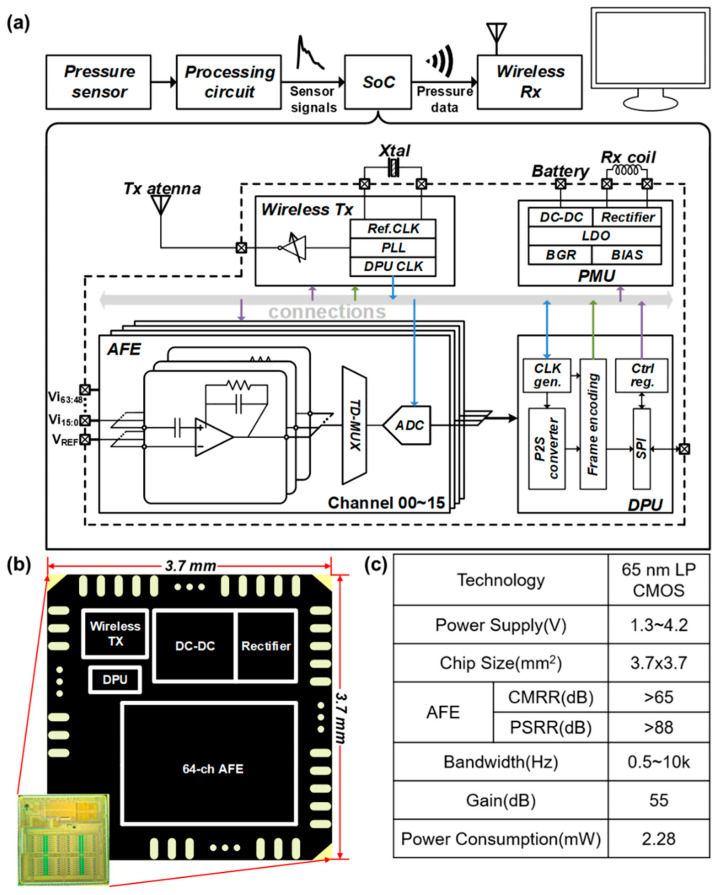
(**a**) The architecture of the 64-channel wireless signal readout SoC. (**b**) Micrograph of the SoC fabricated in 65 nm CMOS process. (**c**) Key specifications of the SoC.

**Figure 5 sensors-22-03934-f005:**
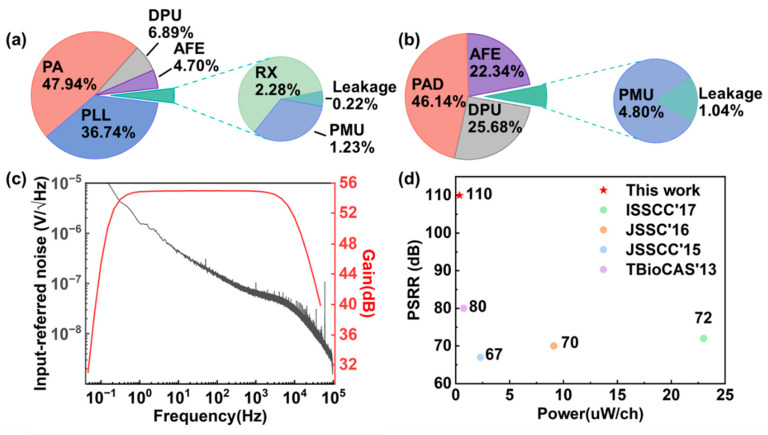
Measured performance of the custom-designed chip. (**a**) Power distribution in wireless mode with a total power of 2.28 mW. (**b**) Power distribution in wireline mode with a total power of 0.48 mW. (**c**) The measured transfer characteristic and input-referred noise of a typical AFE channel. (**d**) Comparison of the PSRR and power consumption of the signal-channel AFE.

**Figure 6 sensors-22-03934-f006:**
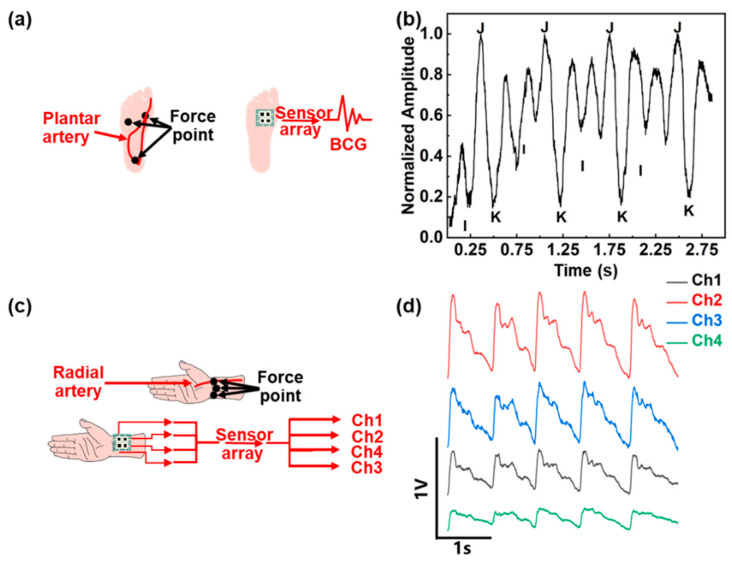
Monitoring of the BCG and pulse wave signals by the proposed system. (**a**) Diagram of the hardware setup to acquire BCG and (**b**) the measured waveform. (**c**) Diagram of the hardware setup to acquire four-channel pulse wave and (**d**) the measured waveforms.

## Data Availability

The data presented in this study are available on request from the corresponding author.

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
