# Peer review of "Flexible Pressure Sensor Array with Multi-Channel Wireless Readout Chip"

_sensors, 2022, doi:10.3390/s22103934_

Round 1
Reviewer 1 Report
The present manuscript deals with flexible pressure sensor arrays used in wearable physiological signal recording applications. It is basically clear, and the main conclusions supported by the data presented. However, I would suggest the authors to provide more descriptions on the experimental details and material features. Besides, a few comments need to be addressed before it can be accepted for publication.
- The preparation method and detailed introduction of sensor materials are not given in this paper. The sensor array in this paper can achieve micro pressure resolution and fast resolution. Is it mainly due to the material or the structure of the system? Can the technology be carried out on different materials with the same performance?
- The sensor array in this paper is mainly used in flexible patch. How about the flexibility and robustness of the arrays? When testing the patch on the soles of the feet and wrists of the human body, is there any difference in signal caused by bending?
- Is there crosstalk between different devices when testing with arrays? If the crosstalk effect of small signals is present, the signal difference in Figure 6 (d) cannot be accurately used to calculate the PTT and PWV.
- Some words in Figure 3 (a) and Figure 4 (a) cannot be clearly recognized.
- The back color of Figure 5 is black, which should be modified.
Author Response
Thank you for your efforts in reviewing our submission. We have carefully reviewed the comments and have revised the manuscript accordingly and thoroughly. Our responses are summarized in a point-by-point manner as below.
Q1. The preparation method and detailed introduction of sensor materials are not given in this paper. The sensor array in this paper can achieve micro pressure resolution and fast resolution. Is it mainly due to the material or the structure of the system? Can the technology be carried out on different materials with the same performance?
A1. The sensors used in this paper are commercially available. To make it clear, we have added the following contents in session 2:
“Figure 1 shows the diagram of the proposed remote health monitoring system, which includes a flexible pressure sensor array (RP-C7.6LT-LF2, Legact Technology co. ltd.) and a custom-designed wireless multi-channel voltage readout chip.”
The micro pressure resolution and fast response are mainly due to the material. Figure 2 shows the tested performance of the sensor. We have revised session 2 to emphasize this:
“The tensile machine applies a time-varying force to the sensor and the electrochemical workstation measures its resistance change over time. The measured performance without post-processing shows the intrinsic characteristics of the sensor, which are shown in Figure 2(b-d).”
In the application, the sensitivity of the system also benefits from the proposed non-linear resistance readout system, which can be applied to sensors with similar inversely proportional resistance responses. To clarify this, we have added the following contents to the end of the discussion part (section 4):
“The proposed non-linear readout scheme is not limited to the specific material used in the paper, but also can be applied to other resistive sensors with similar inversely proportional responses.”
Q2. The sensor array in this paper is mainly used in flexible patch. How about the flexibility and robustness of the arrays? When testing the patch on the soles of the feet and wrists of the human body, is there any difference in signal caused by bending?
A2. The flexibility and robustness of flexible sensors are essential in wearable devices. During the test of the sensor, we repeatedly bent the sensor. Under a certain degree of bending, the sensor can still work typically. For each sensor, the curvature of the human foot and wrist (at the radial artery) is minimal. The difference in the signal caused by bending is negligible. We add the following contents to the discussion part(section 4):
“The robustness of physiological signal recording in a moving human body is important in wearable devices. As the size of each sensor used in this paper is small compared to the curvature radius of the wrist and heel, the bending of the foot and the wrist causes little sensor deformation. Meanwhile, the good bending reliability of the flexible sensor leads to little signal distortion.”
Q3. Is there crosstalk between different devices when testing with arrays? If the crosstalk effect of small signals is present, the signal difference in Figure 6 (d) cannot be accurately used to calculate the PTT and PWV.
A3. To avoid the crosstalk effect between different devices, each sensor path is controlled by an independent switch. As shown in Figure 3(a), when one sensor path turns on, there is no voltage on the other sensor paths to avoid crosstalk between sensors. To clarify this, we have added the following contents to the first paragraph about figure 3(a) in section 2:
“Each node in the sensor array is independently connected to a switch and a divider resistor to avoid crosstalk between different channels. Only the sensor node being measured is turned on to generate a pressure-depended output voltage signal, while the others are disconnected from the bias voltage.”
Q4. Some words in Figure 3 (a) and Figure 4 (a) cannot be clearly recognized. The back color of Figure 5 is black, which should be modified.
A4. We have redrawn all the figures for clear display.
1) In Figure 3(a) and 4(a), we have changed the font and increased the dpi.
2) In Figure 5, the background of the image has been changed to white.
Reviewer 2 Report
The paper presents a highly-integrated remote health monitoring system integrating a flexible pressure sensor array with a multi-channel wireless readout chip. Overall, the topic is very interesting and the integration of flexible pressure sensor array and wireless readout chip is novel. The design of the experiment is sound as well as the results and writing. In my opinion, this paper is fine to be published. But here are some minor problems:
1. The resolution of Fig4a is not very high.
2. On the left of Fig5, the serial number of two figures is missing.
3. About Fig2d, there are not any explanations about delay which may be interesting to readers.
Author Response
Thank you for your efforts in reviewing our submission. We have carefully reviewed the comments and have revised the manuscript accordingly and thoroughly. Our responses are summarized in a point-by-point manner.
Q1. The resolution of Fig4a is not very high. On the left of Fig5, the serial number of two figures is missing.
A1. We have revised all the figures to solve the display problem:
- In Figure 4(a), we have changed the font and increased the dpi.
- In Figure 5, the background of the image has been changed to white and the missing numbers have been added.
Q2. About Fig2d, there are not any explanations about delay which may be interesting to readers.
A2. To explain this, we add the following contents to the end of the paragraph about figure 2d in section 2:
“The current responses immediately to the pressure change almost with no delay, which reveals good repeatability at a wide range of pressure. The lower delay time leads to faster response, which can be used to track pressure changes induced by physiological signals.”
Reviewer 3 Report
The article “Flexible Pressure Sensor Array with Multi-Channel Wireless Readout Chip” describes the preparation of a “remote health monitoring system integrating a flexible pressure sensor array with a multi-channel wireless readout chip” to measure the ballistocardiogram (BCG) and pulse wave signals. This is a low power device used for health monitoring signals, namely BCG and pulse waves during sitting, sleep, etc. The article is written in good English, however some little errors occurred, which corrections are suggested bellow. The article is submitted in the framework of a special issue and is fitting the topic, however there is a concern with the organization of this manuscript, the technical description of the device and procedure occupies most of the manuscript (4 ½ pages), while there are only ~2 pages of results (application). Not sure is the most appropriate construction. It is suggested to review this aspect.
Page 1, Abstract: Please define the abbreviation for “ballistocardiogram” as “BCG” here in order to be further used, e.g. in Introduction.
Page 1, Introduction: The authors mention the piezoresistive presure sensors, however is not clearly stated that this is the case of the sensor they prepared. The authors mention 2 studies with presure sensors for human disease monitoring, ref 16 and 17, are these sensors piozoresistive? Is not clear at all from the Introduction what is the state of the art for piezorezistive sensors and what is the novelty of the sensor developed here. The introduction has to be greatly improved.
Page 1, last paragraph: Is not usual this kind of description in articles, this is more for thesis, not sure if makes sense here. Better remove.
Page 2, last sentence : “The current modifications immediately to pressure almost without delay”. There is a verb missing in this sentence, it does not make sense. Please check.
Page 6: This sentence needs to be changed, since is not clear what was meant to say “High density sensor array requires the signal readout to be compatible with multiple channels has been set up“, maybe “High density sensor array that requires the signal readout to be compatible with multiple channels has been set up”?
Author Response
Thank you for your efforts in reviewing our submission. We have carefully reviewed the comments and have revised the manuscript accordingly and thoroughly. Our responses are summarized in a point-by-point manner. Please see the attachment for details.

Reviewer 4 Report
The authors have presented a great effort in recording BCG signals with their flexible pressure sensor array SOC. Different configurations have been discussed in the methods section, but the outcome of the presented application is not clear in the results. The sensor has been previously reported in their work and is now used in BCG recording in five subjects. Did the authors report the results for PW as outlined in their abstract? It would be interesting to expand the study with more subjects. I understand that this work has high merit for translating into the clinical setting. IRB number is required in the method section. BCG recordings should be quantified in five subjects. What do these amplitudes mean to clinicians in different positions? What would be the best position to record these amplitudes? Significant variations need to be highlighted based on the discussion section's various settings/ positions. Including more subjects would add more clinical value to their proposed work. I would reconsider this work.
Author Response

(The authors gave the same response as above.)

Round 2
Reviewer 1 Report
I'm happy to see that the manuscript has been carefully revised. Most of my comments has been addressed by the authors. It is basically sound and acceptable. I think the manuscript can be accepted in present form.
Reviewer 4 Report
The authors have addressed the concerns.